# Pathway Analysis of a Transcriptome and Metabolite Profile to Elucidate a Compensatory Mechanism for Taurine Deficiency in the Heart of Taurine Transporter Knockout Mice

**Takashi Ito [1],\*** [ID]**, Shigeru Murakami [1] and Stephen Schaffer [2]**

1   Faculty of Biotechnology, Fukui Prefectural University, Fukui 910-1195, Japan; murakami@fpu.ac.jp
2   College of Medicine, University of South Alabama, Mobile, AL 36688, USA; sschaffe@southalabama.edu
*   Correspondence: tito@fpu.ac.jp; Tel.: +81-776-61-6000

**Abstract:** Taurine, which is abundant in mammalian tissues, especially in the heart, is essential for cellular osmoregulation. We previously reported that taurine deficiency leads to changes in the levels of several metabolites, suggesting that alterations in those metabolites might compensate in part for tissue taurine loss, a process that would be important in maintaining cardiac homeostasis. In this study, we investigated the molecular basis for changes in the metabolite profile of a taurine-deficient heart using pathway analysis based on the transcriptome and metabolome profile in the hearts of taurine transporter knockout mice (TauTKO mice), which have been reported by us. First, the genes associated with transport activity, such as the solute carrier (SLC) family, are increased in TauTKO mice, while the established transporters for metabolites that are elevated in the TauTKO heart, such as betaine and carnitine, are not altered by taurine deficiency. Second, the integrated analysis using transcriptome and metabolome data revealed significant increases and/or decreases in the genes involved in Arginine metabolism, Ketone body degradation, Glycerophospholipid metabolism, and Fatty acid metabolism in the KEGG pathway database. In conclusion, these pathway analyses revealed genetic compensatory mechanisms involved in the control of the metabolome profile of the taurine-deficient heart.

**Keywords:** taurine; osmoregulation; metabolomics; transcriptomics; pathway analysis

## 1. Introduction

Taurine is abundant in mammalian tissue, especially in excitable tissues, such as the heart. Taurine functions as a compatible organic osmolyte, thereby assisting in the regulation of intracellular osmotic balance as well as betaine (also called glycine betaine), glycerophosphocholine (GPC), sorbitol, free amino acids, etc. [1]. Taurine also possesses various cellular actions, such as modulation of ion movement and calcium handling [2]. Maintenance of certain species, such as cats, on a taurine-deficient diet leads to development of dilated cardiomyopathy [3]. We previously demonstrated that knocking out the taurine transporter (TauT; Slc6a6) of mice (TauTKO mice) also caused a taurine-deficient cardiomyopathy characterized by ventricular wall thinning and induction of heart failure marker genes [4]. Despite the severe depletion of taurine (less than 1% of wild-type mice), cardiac output is normal in the young animal although it declines with age. Moreover, the decrease in longevity is not severe (median lifespan is about 18 months) [4–6]. The mild condition of the TauTKO phenotype may relate to homeostatic mechanisms that compensate for detrimental effects caused by taurine depletion.

Warskulat et al. reported that some amino acids, including alanine, glutamine, and glutamate, accumulate in the heart of TauTKO mice [7], which may be caused by a compensatory event associated

with taurine loss. We previously performed LC-MS-based metabolome analysis in TauTKO mice to comprehensively analyze the changes in metabolites [8] and observed increases in organic osmolytes, such as betaine, GPC, and amino acids. These observations imply the importance of metabolic control of organic osmolytes in providing protection to the stressed heart.

In the present study, we investigated the genetic mechanisms involved in taurine-depletion-induced metabolome changes of the TauTKO mouse that could contribute to compensation for disturbances of cellular osmoregulation.

## 2. Materials and Methods

*Pathway Analysis*

The online databases DAVID 6.8 (http://david.ncifcrf.gov/home.jsp) [9] and MetaboAnalyst 4.0 (http://www.metaboanalyst.ca/MetaboAnalyst/faces/home.xhtml) [10] were employed for the enrichment and pathway analysis. The transcriptome profile and metabolite profile data used in this analysis have been reported previously [8,11]. For transcriptome analysis, hearts isolated from 3-month-old male TauTKO mice and wild-type mice (*n* = 3) were subjected to cDNA microarray (SurePrint G3 Mouse Gene Expression 8 × 60 K arrays (Agilent Technologies, Santa Clara, CA, USA)) [11]. For metabolome analysis, hearts isolated from 3-month-old male TauTKO mice and wild-type mice (*n* = 4) were subjected to LC-MS-based metabolome analysis [8]. The transcriptome data set used in the pathway analyses consists of the genes that change more than 1.5-fold and have a False Discovery Rate (FDR) that is less than 0.1. The metabolome data set used in the analysis consists of the metabolites that have a variable importance in projection (VIP) score calculated by partial least squares discriminant analysis (PLS-DA) greater than 1 (reported in reference 8) and/or a *p* value calculated by Mann–Whitney U test less than 0.05. A list of the metabolites used in the pathway analysis is shown in Table 1. Since increases in acyl-carnitine and acetyl-carnitine content of the heart may be related to the accumulation of acyl-coenzyme A (CoA) and acetyl-CoA, these were listed as they also reflect changes in acyl-carnitine. For the metabolites that were undetectable in either TauTKO heart or wild-type heart in the previous metabolome analysis, fold-change (FC) values were filled with 10 (increased in TauTKO) or −10 (decreased in TauTKO) for data analysis since the highest value of fold change is 9.2 (Betaine).

**Table 1.** Metabolites significantly changed in the heart of taurine transporter knockout (TauTKO) mice compared to wild-type mice.

| Metabolite | HMDB | KEGG | FC |
|---|---|---|---|
| S-Adenosylmethionine | HMDB0001185 | C00019 | >10 [*1] |
| Argininosuccinate | HMDB00052 | C03406 | >10 [*1] |
| L-Pyroglutamate | HMDB00267 | C01879 | >10 [*1] |
| Betaine | HMDB00043 | C00719 | 9.2 |
| Stachydrine | HMDB04827 | C10172 | 5.9 |
| Ornitine | HMDB0000214 | C00077 | 5.4 |
| L-Histidine | HMDB00177 | C00135 | 3.9 |
| L-Homocarnosine | HMDB00745 | C00884 | 3.7 |
| L-Serine | HMDB00187 | C00065 | 3.3 |
| Acetylcholine | HMDB00895 | C01996 | 3.2 |
| L-Asparagine | HMDB00168 | C00152 | 3.1 |
| L-Citrulline | HMDB00904 | C00327 | 2.9 |
| Acetylcarnitine | HMDB00201 | C02571 | 2.8 |
| L-Glutamine | HMDB00641 | C00064 | 2.7 |
| Acyl (8:0)-carnitine | HMDB0000791 | C02838 | 2.4 |
| Acetylcarnitine | HMDB00201 | C02571 | 2.3 |
| Glycerophosphocholine | HMDB00086 | C00670 | 2.3 |
| Porphobilinogen | HMDB00245 | C00931 | 2.2 |

**Table 1.** *Cont.*

| Metabolite | HMDB | KEGG | FC |
|---|---|---|---|
| Isocitrate | HMDB00193 | C00311 | 2.2 |
| Butylcarnitine | HMDB00201 | C02571 | 2.1 |
| Hexanylcarnitine | HMDB0000756 | | 2.1 |
| L-Carnitine | HMDB00062 | C00318 | 2 |
| Propioylcarnitine | HMDB0000824 | C03017 | 1.9 |
| Tyrosine | HMDB0000158 | C00082 | 1.7 |
| Adenosine | HMDB0000050 | C00212 | −1.4 |
| CDP-choline | HMDB0001413 | C00307 | −1.7 |
| Glycerol 1-phosphate | HMDB00126 | C00093 | −2.4 |
| Glutaurine | HMDB0004195 | C05844 | $< -10$ [*1] |
| Taurine | HMDB00251 | C00245 | $< -10$ [*1] |
| *Carnitine form → Coenzyme A (CoA) form* | | | |
| Acetyl-CoA | HMDB0001206 | C00024 | 2.8 |
| Octanoyl-CoA | HMDB0001070 | C01944 | 2.4 |
| Hexanoyl-CoA | HMDB0002845 | C05270 | 2.1 |
| Butanoyl-CoA | HMDB0001088 | C00136 | 2.1 |
| Propionyl-CoA | HMDB0001275 | C00100 | 1.9 |

[*1] These values were filled with 10 or −10 for data analysis, since these were missing values due to being undetectable in either TauTKO mice or wild-type. KEGG, Kyoto Encyclopedia of Genes and Genomes; FC, fold-change.

The joint pathway analysis tool of MetaboAnalyst 4.0 was used for the integrated pathway analysis. Both the datasets of the differentially expressed genes and the significantly changed metabolites were uploaded in the meantime to this database. Fisher's exact test was chosen for the over-representation analysis and Degree Centrality was chosen for the topology analysis. The integrated pathway analysis tool gave the list of enriched pathways and respective KEGG maps.

## 3. Results

### 3.1. Functional Annotation Analysis in the Transcriptome Profile of Taurine-Depleted Heart

We previously demonstrated by microarray analysis differentially expressed genes in the transcriptome profile between TauTKO and wild-type mice [11]. In the present study, to identify genes which are involved in metabolite changes within the TauTKO heart, we performed functional annotation analysis using the online functional annotation tool DAVID to identify Gene Ontology (GO) clusters significantly over-represented in the differentially expressed genes (Tables 2–6). Concerning metabolism, GO molecular function terms for transport activity, including symporter activity and amino acid transmembrane transporter activity, are enriched in genes of the TauTKO mice, whereas the term for metabolic activity, including catabolic activity, is enriched in genes that were decreased.

**Table 2.** Gene Ontology (GO) terms enriched in the increased gene set in TauTKO mice.

| Term | Count | % | *p* Value | Benjamini |
|---|---|---|---|---|
| GO:0005515~protein binding | 125 | 25.7 | $1.39 \times 10^{-5}$ | 0.008592 |
| GO:0008017~microtubule binding | 15 | 3.1 | $1.16 \times 10^{-4}$ | 0.035231 |
| GO:0015293~symporter activity | 11 | 2.3 | $1.54 \times 10^{-4}$ | 0.031392 |
| GO:0005178~integrin binding | 9 | 1.9 | 0.001434 | 0.199441 |
| GO:0008201~heparin binding | 11 | 2.3 | 0.001632 | 0.183368 |
| GO:0005509~calcium ion binding | 28 | 5.8 | 0.002558 | 0.232527 |
| GO:0030676~Rac guanyl-nucleotide exchange factor activity | 4 | 0.8 | 0.003029 | 0.235644 |
| GO:0015171~amino acid transmembrane transporter activity | 5 | 1.0 | 0.005909 | 0.368297 |
| GO:0005215~transporter activity | 12 | 2.5 | 0.007494 | 0.404414 |
| GO:0005328~neurotransmitter:sodium symporter activity | 4 | 0.8 | 0.008623 | 0.415453 |

**Table 3.** Increased genes in GO:0015293~symporter activity.

| Gene Symbol | Gene Name |
| --- | --- |
| Slc1a2 | solute carrier family 1 (glial high affinity glutamate transporter), member 2 |
| Slc10a7 | solute carrier family 10 (sodium/bile acid cotransporter family), member 7 |
| Slc16a12 | solute carrier family 16 (monocarboxylic acid transporters), member 12 |
| Slc17a7 | solute carrier family 17 (sodium-dependent inorganic phosphate cotransporter), member 7 |
| Slc24a5 | solute carrier family 24, member 5 |
| Slc38a2 | solute carrier family 38, member 2 |
| Slc38a4 | solute carrier family 38, member 4 |
| Slc6a17 | solute carrier family 6 (neurotransmitter transporter), member 17 |
| Slc6a1 | solute carrier family 6 (neurotransmitter transporter, gamma-aminobutyric acid (GABA), member 1 |
| Slc6a9 | solute carrier family 6 (neurotransmitter transporter, glycine), member 9 |
| Slc6a6 | solute carrier family 6 (neurotransmitter transporter, taurine), member 6 |

**Table 4.** Increased genes in GO:0015171~amino acid transmembrane transporter activity.

| Gene Symbol | Gene Name |
| --- | --- |
| Pdpn | podoplanin |
| Slc38a2 | solute carrier family 38, member 2 |
| Slc38a4 | solute carrier family 38, member 4 |
| Slc6a17 | solute carrier family 6 (neurotransmitter transporter), member 17 |
| Slc7a4 | solute carrier family 6 (neurotransmitter transporter, GABA), member 1 |

**Table 5.** GO terms enriched in the decreased gene set in TauTKO mice.

| Term | Count | % | *p* Value | Benjamini |
| --- | --- | --- | --- | --- |
| GO:0004364~glutathione transferase activity | 6 | 1.9 | 0.000049 | 0.018 |
| GO:0005244~voltage-gated ion channel activity | 8 | 2.5 | 0.0014 | 0.23 |
| GO:0003824~catalytic activity | 15 | 4.7 | 0.0025 | 0.28 |

**Table 6.** Decreased genes in GO:0003824~catalytic activity.

| Gene Symbol | Gene Name |
| --- | --- |
| Bdh1 | 3-hydroxybutyrate dehydrogenase, type 1 |
| Abat | 4-aminobutyrate aminotransferase |
| Pfkfb1 | 6-phosphofructo-2-kinase/fructose-2,6-biphosphatase 1 |
| Cdk5rap1 | CDK5 regulatory subunit associated protein 1 |
| Acat1 | acetyl-Coenzyme A acetyltransferase 1 |
| Acaa2 | acetyl-Coenzyme A acyltransferase 2 |
| Aldob | aldolase B, fructose-bisphosphate |
| Eci1 | enoyl-Coenzyme A delta isomerase 1 |
| Fah | fumarylacetoacetate hydrolase |
| Gpt | glutamic pyruvic transaminase |
| Isoc2a | isochorismatase domain containing 2a |
| Klhl3 | kelch-like 3 |
| Ldhd | lactate dehydrogenase D |
| Odc1 | ornithine decarboxylase, structural 1 |
| Ppm1h | protein phosphatase 1H (PP2C domain containing) |

Transporters

Our previous metabolome analysis revealed that some organic osmolytes are increased in TauTKO mice [11]. M-A plots of transcriptome data are shown in Figure 1, and the transporter genes that were either elevated in TauTKO mice or relate to the focus of this study are highlighted. Among the induced transporters are Slc38a2 and Slc38a4, which represent amino acid transporter system A-2 (ATA2) and A-3 (ATA3), respectively. The induction of these transporters has been observed in

previous reports [5,8]. However, the amino acid transporters Slc6a9 (glycine transporter-1) and Slc7a4 (cationic amino acid transporter-4) were also increased in the TauTKO mouse. Although both Slc6a1 (gamma-aminobutyric acid (GABA) transporter-1) and Slc6a17 (its substrate is unidentified) are also elevated, their expression remains low. Additionally, the truncated mRNA of Slc6a6 (taurine transporter) was also detected by microarray analysis in TauTKO tissue.

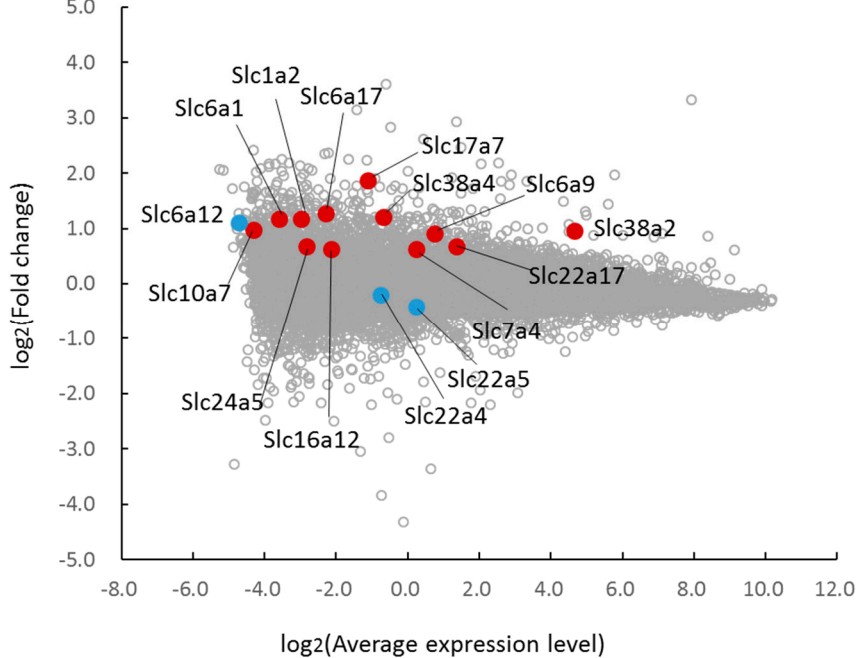

**Figure 1.** Changes in transporter genes of TauTKO mice. M-A plot showing an intensity-dependent ratio of raw gene microarray data. The genes that were significantly altered and of special interest are highlighted in red and light blue, respectively. Expression levels were normalized by 75th percentile genes for each sample (75th percentile = 0 in log (expression level)).

Whereas betaine/GABA transporter-1 (BGT-1; Slc6a12), which is responsible for betaine uptake, is induced/activated by hyperosmotic conditions in certain cells, expression of BGT-1 is low in the heart and not significantly different between TauTKO and wild-type mice.

Moreover, expression of the organic cation transporters OCTN1 (Slc22a4) and OCTN2 (Slc22a5), specific transporters for carnitine, is not different between TauTKO and wild-type mice. By contrast, the expression of the other Slc22 family member, Slc22a17 (its substrate is unidentified [12]), is increased.

*3.2. Integrated Pathway Analysis of Transcriptome and Metabolite Data*

We also investigated changes in the metabolic pathways of the TauTKO mouse by integrated pathway analysis of transcriptome and metabolomics data of the TauTKO mouse using the online database MetaboAnalyst 4.0. The analysis of differentially expressed genes and differentially contained metabolites revealed alterations in some KEGG (Kyoto Encyclopedia of Genes and Genomes) metabolic pathways (Table 7). No pathways showed statistically significant hits scores ($p < 0.05$), whereas seven pathways showed high hits scores (7–11), which appear more than expected. Since the KEGG pathway includes many genes which are not expressed in the heart, it may be biased to identify the activated pathway based on $p$-value. The genes and metabolites that were significantly different are highlighted on the KEGG metabolic pathway maps (Figures 2–5).

**Table 7.** Activated pathways in heart suggested from the integrated pathway analysis.

| Pathway | Total | Expected | Hits | *p* Value | Topology |
|---|---|---|---|---|---|
| **Arginine and proline metabolism** | 102 | 6.4818 | 11 | 0.052661 | 0.43011 |
| **Synthesis and degradation of ketone bodies** | 15 | 0.95321 | 3 | 0.06475 | 1.3333 |
| **Fatty acid elongation** | 58 | 3.6858 | 7 | 0.069782 | 0.47945 |
| **Alanine, aspartate, and glutamate metabolism** | 58 | 3.6858 | 7 | 0.069782 | 0.4898 |
| **Fatty acid metabolism** | 88 | 5.5922 | 8 | 0.18998 | 0.99029 |
| **Glutathione metabolism** | 79 | 5.0203 | 7 | 0.23141 | 0.16981 |
| **Valine, leucine and isoleucine degradation** | 88 | 5.5922 | 7 | 0.32376 | 0.25556 |
| **Glycerophospholipid metabolism** | 119 | 7.5622 | 8 | 0.49038 | 0.26667 |

### 3.2.1. Glycerophospholipid Metabolism

Since GPC is one of the organic osmolytes that increases under hyperosmotic conditions, we focused on its synthetic pathway. As shown in Figure 2, some metabolites and some genes involved in glycerophospholipid metabolism are altered in TauTKO mice. The observed increase in GPC in the TauTKO mouse heart is in agreement with the elevated expression of phospholipase A2 (Pla2g4a), suggesting that this pathway may be responsible for an increase in GPC. Additionally, acetylcholine cholinesterase (Ache) and CDP-choline are changed, suggesting that the stimulation of the pathway enhances phosphatidylcholine synthesis. Moreover, phosphatidic acid phosphatase type 2C (Ppap2c), which catalyzes the conversion of phosphatidic acid to diacylglycerol, is elevated in TauTKO mice, indicating that this activation contributes to enhanced phosphatidylcholine synthesis. A reduction in glycerol-1-phosphate may be caused by enhancement of diacylglycerol synthesis.

### 3.2.2. Arginine and Proline Metabolism and Alanine, Aspartate, and Glutamate Metabolism

As shown in Figure 3A (Arginine and proline metabolism pathway), the induction of arginosccinate lyase (Asl) occurs concomitantly with increases in Ornitine, Citrulline, and Arginosuccinate in the TauTKO heart. The reason why arginosuccinate is increased despite the induction in the Asl gene may be the increase in citrulline and ornithine in the heart. Most of the genes involved in the urea cycle are expressed in the liver. In the case of other tissues, Asl catalyzes the formation of arginine from arginosuccinate. Arginine is a precursor of nitric monoxide (NO), whose formation is catalyzed by NO synthase (NOS) [13]. It has been reported that Asl is a key enzyme in the production of NO in the heart [14]. Asl and arginosuccinate overlap with Alanine, aspartate, and glutamate metabolism (Figure 3B); this map indicates the importance of asparagine in supplying arginosuccinate. These data indicate that the arginine metabolic pathway involving NOS generation may be activated in the heart of TauTKO mice.

### 3.2.3. Fatty Acid Metabolism and Degradation of Ketone Bodies

As shown in Figure 4A, the genes of the fatty acid metabolic pathway, Acat and Acaa, are significantly decreased. In addition to these genes, Ehhadh, Hadh, Hadha, and Echs1 are also slightly diminished in TauTKO mice (by −1.3~−1.4-fold). According to the metabolic profile reported previously, short-chain acylcarnitine (Butylcarnitine (C4), Hexanoylcarnitine (C6), and octanoylcarnitine (C8)) and acetylcarnitine (C2) are higher in TauTKO mice than in wild-type mice [8]. Since carnitine plays a role in the uptake of fatty acids by the mitochondria where they are converted to acyl-CoA and undergo oxidation, an increase in acyl-carnitine may result from incomplete fatty acid oxidation. Moreover, carnitine conjugation is responsible for the detoxification of excess acyl-CoAs [15]. For example, butyrylcarnitine is increased in the plasma of the patients of Short-chain acyl-CoA dehydrogenase deficiency [16]. Therefore, these data suggest that oxidation of short chain acyl-CoAs is suppressed in TauTKO mice.

Additionally, as shown in Figure 4B, the genes of the ketone body degradation pathway, Bdh1 and Acot, are decreased in TauT KO mice, suggesting that the availability of ketone bodies in the heart is reduced.

### 3.2.4. Valine, Leucine, and Isoleucine Degradation

As shown in Figure 5, the genes of the valine, leucine, and isoleucine degradation pathways, such as Mccc1, Acat1, Abat1, Acaa1, and Aldh4a1, are decreased in the TauTKO heart. According to the metabolic map, a reduction of Mccc may cause a decrease in the conversion of methylclotonyl-CoA to methylglutaconyl-CoA, enhancing the conversion to hydroxyisovaleryl-CoA. A reduction in Abat and Aldh7a1 may suppress the metabolism of (S)-methylmalonate semialdehyde, resulting in enhanced generation of propionyl-CoA conversion from (S)-methylmalonate semialdehyde.

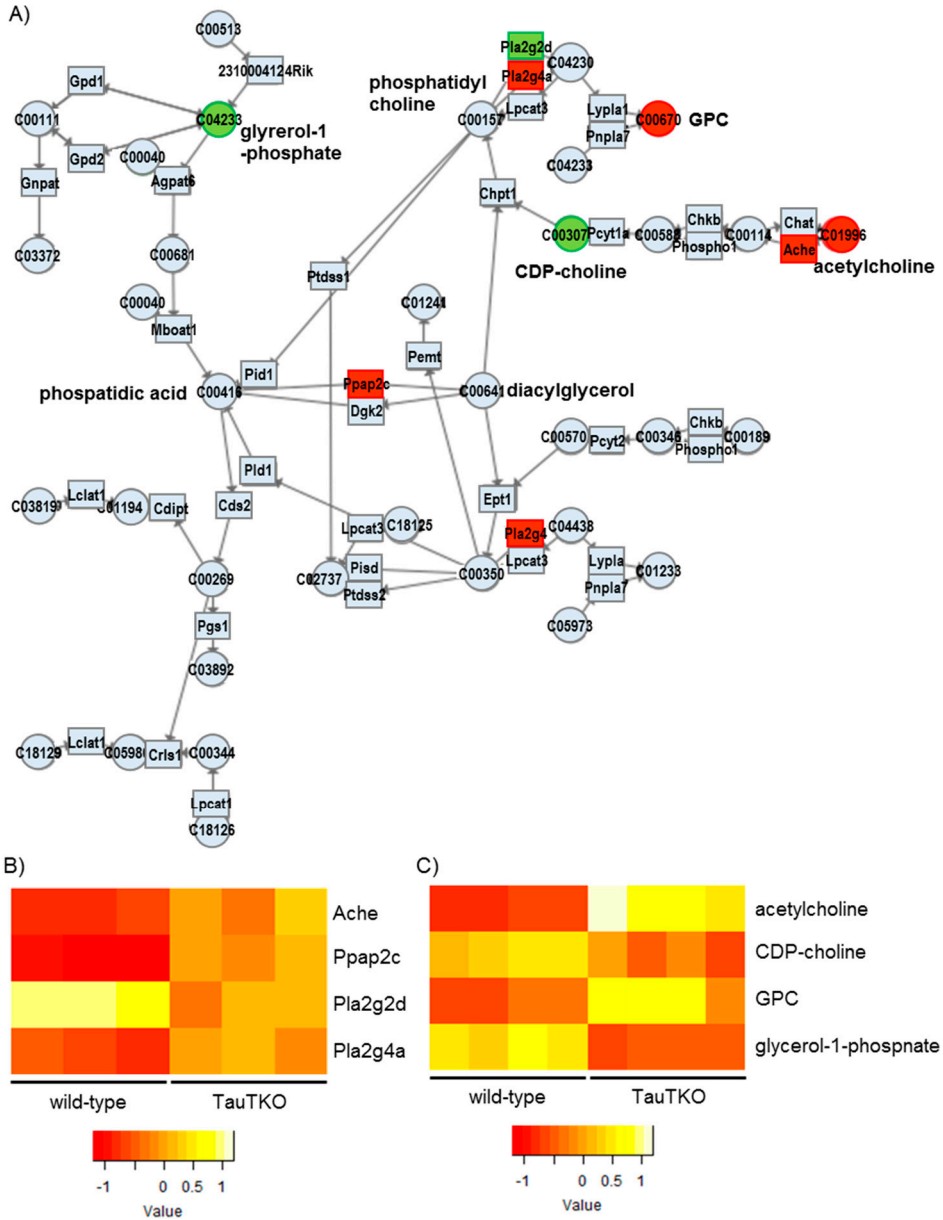

**Figure 2.** Changes in genes and metabolites of the Glycerophospholipid metabolism pathway of TauTKO mice. (**A**) Altered genes and metabolites are highlighted in red for increase and in green for decrease. Numbers in brackets indicate fold-change of genes. (**B**,**C**) Relative levels of the enriched genes (**B**) and metabolites (**C**) are shown. The heat map represents log 2 fold-changes (normalized by the average of all samples) in wild-type and TauTKO mice. GPC; glycerophosphocholine, CDP-choline; cytidine 5′-Diphosphocholine.

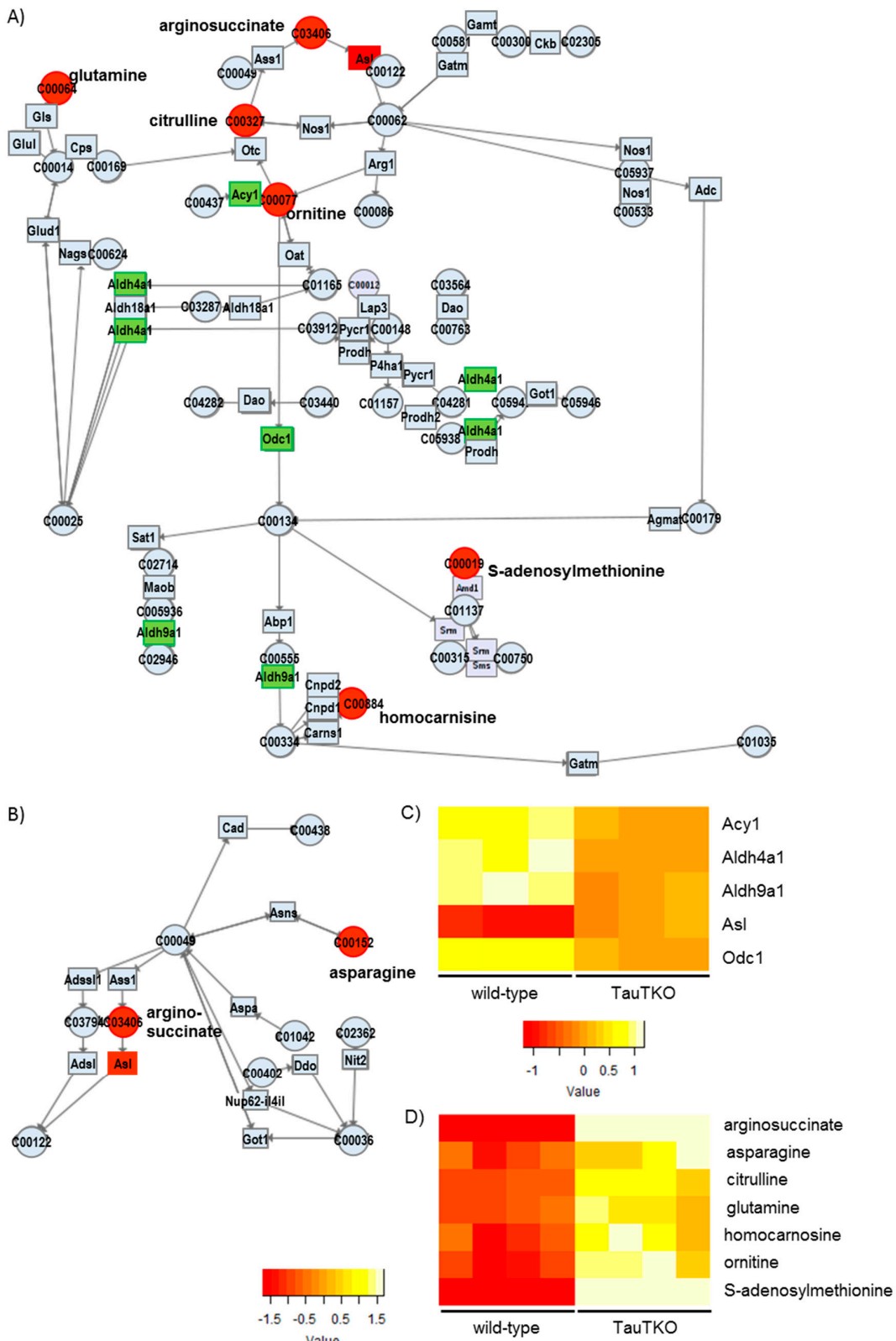

**Figure 3.** Genes and metabolites altered in the Arginine metabolism pathway (**A**) and the Alanine metabolism pathway (**B**) of TauTKO mice. Altered genes and metabolites are highlighted in red for increase and in green for decrease. (**C**,**D**) Relative levels of the enriched genes (**C**) and metabolites (**D**) are shown. The heat map represents log 2 fold-changes (normalized by the average of all samples) in wild-type and TauTKO mice.

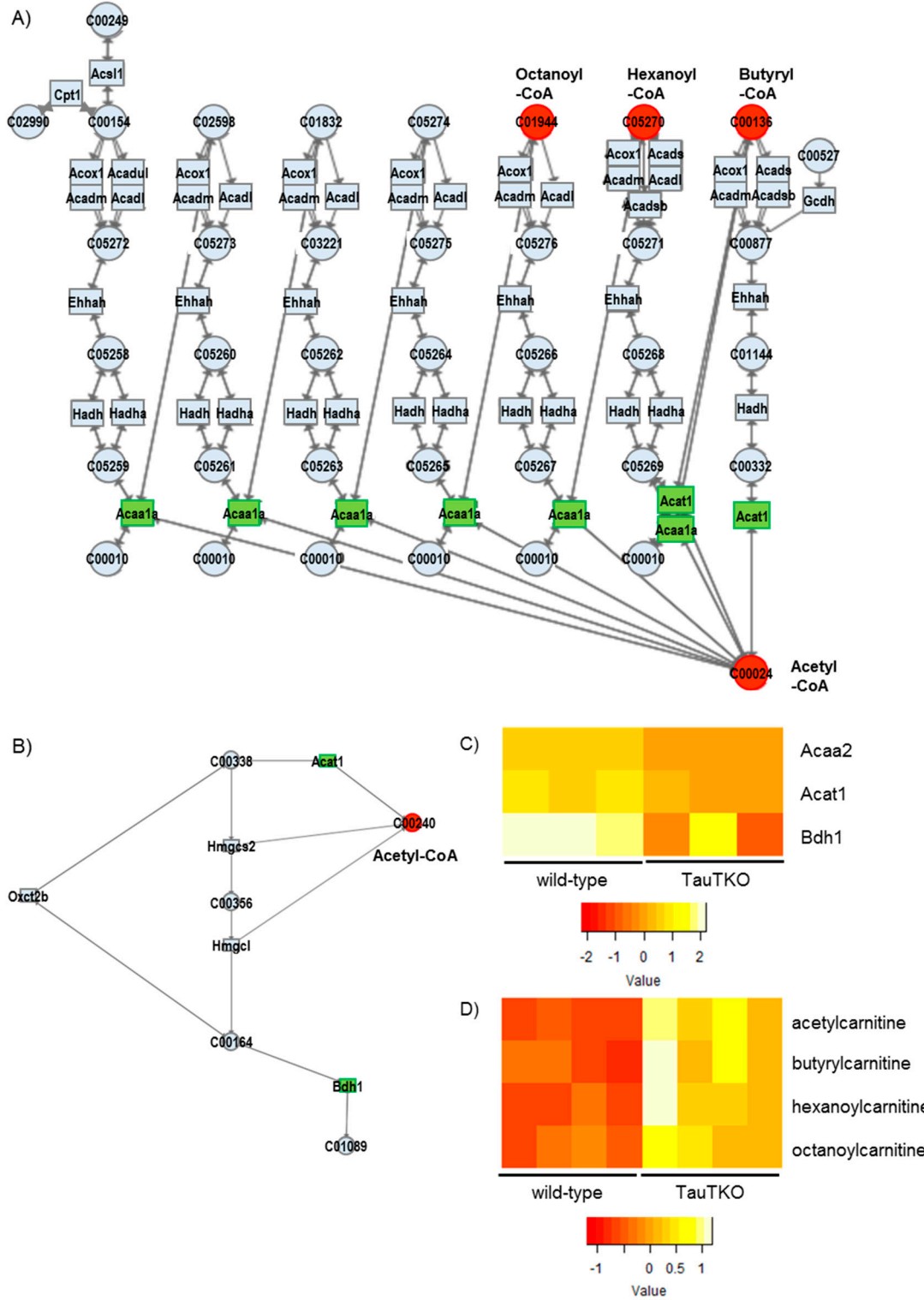

**Figure 4.** Altered genes and metabolites of the fatty acid oxidation pathway (**A**) and the ketone body degradation pathway (**B**) of TauTKO mice. Altered genes and metabolites are highlighted in red for increase and in green for decrease. (**C**,**D**) Relative levels of the enriched genes (**C**) and metabolites (**D**) are shown. The heat map represents log 2 fold-changes (normalized by the average of all samples) in wild-type and TauTKO mice.

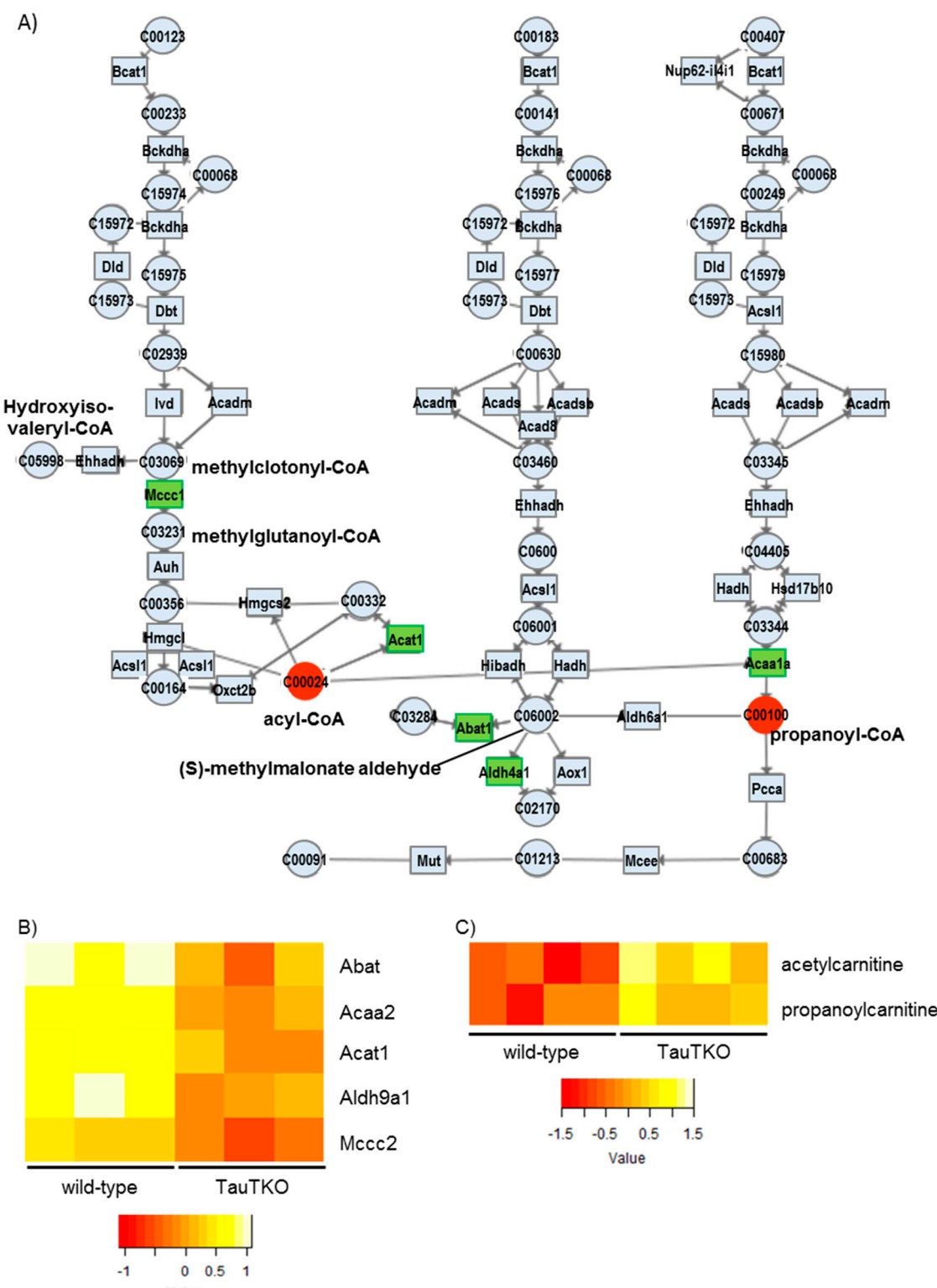

**Figure 5.** Altered genes and metabolites of the branched-chain amino acid (BCAA) metabolism pathway (**A**) in TauTKO mice. Altered genes and metabolites are highlighted in red for increase and in green for decrease. (**B**,**C**) Relative levels of the enriched genes (**B**) and metabolites (**C**) are shown. The heat map represents log 2 fold-changes (normalized by the average of all samples) in wild-type and TauTKO mice.

## 4. Discussion

One of the most important functions of taurine in the cell is the regulation of intracellular osmolality. When cells are exposed to the hyperosmolar milieu, the content of organic osmolytes, including those of taurine, betaine, GPC, and amino acids, are increased, which contributes to the establishment of an ionic balance and minimizes changes in cell volume [1]. In certain types of cells, such as kidney cells, the mRNA of the TauT (Slc6a6), the BGT-1 (Slc6a12), and the amino acid transporter system A (Slc38a2) are increased by hyperosmotic stress, which in turn stimulates the uptake of their respective substrates. These cellular responses against a change in osmolality are regulated by the transcription factor TonEBP (tonicity-response element binding protein; also called NFAT5). In the case of the heart, we observed that taurine depletion (TauT knockout) also alters organic osmolyte content, suggesting that disturbances in cellular osmoregulation caused by the loss of taurine may be compensated for by alterations in other osmolytes. However, the expression of BGT-1 is not significantly different between TauTKO and wild-type mice, implying the involvement of an unidentified mechanism in the change in betaine content. Transcriptome analysis also revealed the induction of other transporters, including Slc38a2, Slc38a4, Slc6a9 (glycine transporter-1), and Slc7a4 (a member of the cationic amino acid transporter y+ system), in the TauTKO heart. In addition to BGT-1, other betaine transport systems exist: Slc6A20 (betaine/proline transporter) and Slc7a6 (another member of a member of the cationic amino acid transporter y+ system, y+ LAT2), which are expressed in mouse oocytes [17,18]. The Slc38 protein family is also a potential transport system for betaine [17]. While these transporters are candidates for betaine transport in the heart, further study is necessary to identify the transporter which contributes to osmoregulation of betaine in the heart.

Carnitine also functions as an organic osmolyte [19]. In the transcriptome profile, levels of the carnitine transporter genes Slc22a4 and Slc22a5 are not altered in TauTKO mice. By contrast, the content of another member of the organic cation transporter family, Slc22a17, is increased in the TauTKO heart; however, carnitine does not function as a substrate of that transporter [12]. Activation of carnitine transport in TauTKO mice may be related to post-translational modifications, such as phosphorylation.

GPC is converted from phosphatidylcholine to lysophosphatidylcholine by Phospholipase A2 and Glycerophosphocholine phosphodiesterase. Alternatively, Phospholipase B (PLB) catalyzes the direct conversion of phosphatidylcholine to GPC [1]. It has been reported that cellular exposure to a hyperosmolar condition increases the content of PLB mRNA, which should mediate an increase in GPC in the kidney cell. In the present study, we observed in the TauTKO heart induction of one of the phospholipase genes, Pla2g4, but not that of the PLB gene (Fold change = 1.243, *p* = 0.237), indicating that a different osmotic-related pathway may function in the heart to control GPC synthesis. In addition, Acetylcholinesterase (Ache) is induced in the TauTKO mouse while acetylcholine level increases and CDP-choline content falls. This pathway may provide the choline portion of GPC's structure. Moreover, phosphatidic acid phosphatase-2c (Ppap2c), which catalyzes the conversion of phosphatidic acid to diacylglycerol, a precursor of phosphatidylcholine, is also induced in the TauTKO mouse. These coordinated activations of the metabolic pathway likely contribute to the effectiveness of GPC as an osmoregulator in the heart.

Besides examining the modulation of various osmolytes in the TauTKO heart, we observed an induction of arginosuccinate lyase, concomitant with an increase in the levels of arginosuccinate, ornithine, and citrulline, in TauTKO mice. Citrulline and arginosuccinate are important intermediates in the production of NO in most tissues, the exception being the liver [13]. The knockout of Asl in mice causes a decrease in protein nitrosylation and nitrite in the heart, evidence of an attenuation of NO synthesis [14]. Therefore, the activation of this pathway in the TauTKO heart may increase NO production.

We observed reductions in some genes associated with fatty acid oxidation in the TauTKO mouse, which may cause incomplete fatty acid oxidation followed by the accumulation of the carnitine derivatives of short chain fatty acids. Furthermore, acetylcarnitine is also higher in TauTKO mice, indicating an accumulation of acetyl-CoA. Additionally, we observed that the gene expressions of

Bdh1 and Acat1 that metabolize to acetyl-CoA from ketone bodies were decreased, but acetyl-CoA was increased. Acetyl-CoA not only feeds carbon into the tricarboxylic acid cycle but can also control energy metabolism by acetylation of lysine residues of key enzymes [20]. Although the enzymes related to fatty acid oxidation, as well as the tricarboxylic acid cycle and the electron transport chain, are targets for acetylation, the effect of acetylation on the activity of fatty acid oxidation is controversial. In the case of the heart, hyperacetylation of mitochondrial proteins caused by a reduction in SIRT3, a mitochondrial NAD-dependent deacetylase, in obesity and diabetes is associated with an increase in the fatty acid oxidation rate [21]. Lysine acetylation can also control transcription of fatty acid oxidation-related genes by modification of transcriptional regulator PGC-1alpha [21]. These acetylation processes, which are mediated by acetyl-CoA accumulation, may slow fatty acid oxidation and ketone degradation in the heart of TauTKO mice. Moreover, the carboxylation of acetyl CoA produces malonyl CoA, which inhibits carnitine palmitoyltransferase, a key enzyme involved in fatty acid uptake and oxidation by the heart. Importantly, similar transcriptome changes related to fatty acid oxidation were observed in skeletal muscle of TauTKO mice [22]. Therefore, this may be a result of tissue taurine depletion. Indeed, the decrease in oxidation by the taurine-deficient rat heart has been largely attributed to taurine-mediated reductions in carnitine palmitoyltransferase-1 activity [23]. Additionally, why acetyl-CoA is increased in the heart of TauTKO mice is unclear. It is possible that the accumulated acetyl-CoA comes from glycolysis pathway activation and/or the reduction of the tricarboxylic acid (TCA) cycle. We previously observed that the glycolysis is enhanced and the TCA cycle activity is diminished in the taurine-deficient heart [23].

Finally, we observed a reduction in some genes associated with the branched-chain amino acid (BCAA) degradation pathways. Catabolic defects of BCAA metabolism occur in heart failure. In the case of the pressure-overload-induced failing heart of mice, most of genes of the KEGG BCAA catabolic pathway are reduced compared to those of the sham-operated heart. Moreover, the genetic changes are accompanied by an increase in the branched-chain keto acids (BCKAs), which are markers for a defect in the BCAA metabolic pathway [24]. Importantly, it has been reported that elevations in BCKAs by knocking out PP2Cm, a BCKA dehydrogenase phosphatase, lead to impaired cardiac function. Similarly, it is attractive to suggest that suppression of the BCAA metabolic pathway in TauTKO mice may partially contribute to the aging-dependent decline in cardiac function.

## 5. Conclusions

In conclusion, the integrated pathway analysis of the transcriptome and metabolome profiles in TauTKO mice identified some metabolic pathways activated by taurine depletion. Some of the changes may be directly caused by taurine depletion, while others may be caused by cardiac disorder. The present study raises the possibility that taurine may play a role in the regulation of GPC metabolism, NO synthesis, fatty acid oxidation, ketone body degradation, and BCAA metabolism in the heart. We may be able to distinguish the direct effects or the secondary effects by comparing with the changes of genes and metabolites in taurine-treated cells/animals. Furthermore, there are very few studies investigating the various osmoregulatory mechanisms in the heart. The present study provides potential molecular pathways responsible for the control of osmolytes, such as betaine, GPC, and amino acids. Further studies are necessary to understand the genetic modulation of osmotic imbalances in the heart.

**Author Contributions:** T.I. conceived and performed the experiments and analyzed the data; T.I., S.M., and S.S. wrote the paper.

**Funding:** This work is supported from the JSPS KAKENHI Grant Numbers 22790097 and 25750368.

**Conflicts of Interest:** The authors declare no conflict of interest.

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
