# Peer review of "Pathway Analysis of a Transcriptome and Metabolite Profile to Elucidate a Compensatory Mechanism for Taurine Deficiency in the Heart of Taurine Transporter Knockout Mice"

_2571-8800, doi:10.3390/j1010007_

Round 1

Reviewer 1 Report

The authors Ito et al. presents us with data regarding transcriptional and metabolite changes in the TauTKO heart providing novel insight into possible changed pathways as well as explaining previous observations. 

Major comments:

The study should be limited to examination of pathways that are found to be significantly altered (or at least tends to be) in the inital analysis. For example, glycerophospholipid metabolism is examined in detail despite this pathway not being significant in Table 5. This part should be removed from both results and discussion.

The relevance of figure 1 can be questioned as the data presented (the names of the transporter genes) are already presented in Table 3 and no further analysis of the M-A plot is used or described. Please remove.

It is not completely clear how the Metaboanalyst analysis were performed, please include a sentence or two on this in the methods section. Especially worrying is the lack of correction for multiple testing in this analysis (table 5), as for table 2 and 4 we get benjamini corrected P-values, but this is not the case for table 5. Please either explain this or provide P-values corrected for multiple testing for table 5.

Author Response

> First, I appreciate the reviewers for his/her very helpful and careful review for our paper. We have revised the manuscript according to the reviewer’s comments. We believe that this paper is improved well. 

The response to the comments is as follows;

(Additional descriptions according to reviewers’ comments are lined by yellow marker (reviewer 1) and light-green marker (reviewer 2), light-blue marker (reviewer 3))

The authors Ito et al. presents us with data regarding transcriptional and metabolite changes in the TauTKO heart providing novel insight into possible changed pathways as well as explaining previous observations.

Major comments:

The study should be limited to examination of pathways that are found to be significantly altered (or at least tends to be) in the inital analysis. For example, glycerophospholipid metabolism is examined in detail despite this pathway not being significant in Table 5. This part should be removed from both results and discussion.

> I agree that statistical significance is important.  However, we could not detect any pathways which shows statistical significance (p<0.05) after we re-analyzed on the revised metabolite list (the revision of metabolite list is suggested by reviewer #3).  Since KEGG pathway include many genes which are not expressed in the heart, it may be bias to identify the activated pathway based on p-value.  Instead, we identified 7 pathways showed high hits scores (7~11) and these scores appear more than the expected scores.  Therefore, we focused some of these pathways in the present study.  This explanation was added in the result section.

The relevance of figure 1 can be questioned as the data presented (the names of the transporter genes) are already presented in Table 3 and no further analysis of the M-A plot is used or described. Please remove.

> M-A plot can show the relative expression level as well as fold change score. I wanted to explain both expression level and fold change of transporter genes by using this M-A plot.  

It is not completely clear how the Metaboanalyst analysis were performed, please include a sentence or two on this in the methods section. Especially worrying is the lack of correction for multiple testing in this analysis (table 5), as for table 2 and 4 we get benjamini corrected P-values, but this is not the case for table 5. Please either explain this or provide P-values corrected for multiple testing for table 5.

> As the reviewer commented, how Metaboanalyst was used was explained in method section.  Fisher’s exact test was chosen for the present study. 

Reviewer 2 Report

This article described the integration of metabolomics and transcriptomics analyses for taurine transport knockout mice. In general, the manuscript was written clearly, and provided plenty of biological insights. However, I have several concerns about the methods that the authors applied:

In Materials and Methods part, the authors only listed pathway analysis, which in my opinion was not enough to present for public audience to obtain reproducible results. Although it is understandable that the experimental setup had been published in the authors' previous papers, the experimental process and setup are still necessary to be briefly described in this part so that the readers can understand and repeat the analyses.

The authors need to rephrase the part about using METLIN to re-analyze metabolomics data. In my best knowledge, METLIN can only give putative annotation for metabolites with accurate mass and fragmentation patterns. What would be the differences between the metabolite analyses conducted in this manuscript and the authors' previously published paper?

Page 2, line 67: for the undetectable metabolites, the authors did not explain why they filled their values with 10 or -10, it is not clear for the readers.

Page 3, line 77-78: what would be the differences between the metabolite annotation method in their former publication and using METLIN in this manuscript?

Page 6, part of "integration pathway analysis": the authors did not describe how they did the integration analysis in detail. For instance, did they upload differentially expressed genes (DEG) and significantly changed metabolites at the meantime, and obtained the enriched pathways and biological processes? OR they upload gene and metabolite lists separately, and then compared the enriched pathways after that?

The authors may plot a heatmap with DEG and significantly changed metabolites to show the different patterns in control and taurine deficient mice group.

Correlation analysis can be a good method for the integration analysis of metabolomics and transcriptomics data.

It is good that the authors pointed out in the conclusion that "some of the changes may be directly caused by taurine depletion, while others may be caused by cardiac disorders". However, how to distinguish the direct effects and the secondary effects? It is understandable that it is beyond the scope of this manuscript to distinguish these two effects by conducting additional experiment. But the authors can provide some possible insights and assumptions.

The authors should give more details how Fig 2-5 were made? Were they made by Cytoscape? It was not stated in the part of Materials and Methods.

The part of metabolite annotation/identification was not satisfying. Because there are four different confidence levels for metabolite annotation. Only with accurate mass for metabolite annotation is the most superficial layer, where the authors must claim this in a very clear way, e.g. state that their annotation was tentative annotations. In addition, the authors still need to clarify the main differences from their last reports. 

Minor comments:

Page 4, line 83: the sentence is broken, not finished.

The second sentence in Abstract has redundant meanings

Abstract, line 23-24: no need to use quotes for pathways

Page 4, line 91-92: the same like above, no need to use quotes for specific biological processes

Author Response

> First, I appreciate the reviewers for his/her very helpful and careful review for our paper. We have revised the manuscript according to the reviewer’s comments. We believe that this paper is improved well. 

The response to the comments is as follows;

(Additional descriptions according to reviewers’ comments are lined by yellow marker (reviewer 1) and light-green marker (reviewer 2), light-blue marker (reviewer 3))

This article described the integration of metabolomics and transcriptomics analyses for taurine transport knockout mice. In general, the manuscript was written clearly, and provided plenty of biological insights. However, I have several concerns about the methods that the authors applied:

In Materials and Methods part, the authors only listed pathway analysis, which in my opinion was not enough to present for public audience to obtain reproducible results. Although it is understandable that the experimental setup had been published in the authors' previous papers, the experimental process and setup are still necessary to be briefly described in this part so that the readers can understand and repeat the analyses.

> As the reviewer#1 and reviewer#2 suggested, I added the explanation of study design of previous study, and how we used Metaboanalyst.  (highlighted in yellow and light-green markers)

The authors need to rephrase the part about using METLIN to re-analyze metabolomics data. In my best knowledge, METLIN can only give putative annotation for metabolites with accurate mass and fragmentation patterns. What would be the differences between the metabolite analyses conducted in this manuscript and the authors' previously published paper?

> I agree that data from METLIN is putative annotation.  Then, we actually could not have confidence that these unknown compounds are Lysophosphatidylcholine, Methylbutanoylcarnitine and Hydroxyisovalerylcarnitine, as the reviewer pointed.  Concerning propionylcarnitine, it was already annotated (I could not notice about it when I prepared the previous manuscript). Since this compound satisfied the criteria (p<0.05), it is listed in Table 1. 

Page 2, line 67: for the undetectable metabolites, the authors did not explain why they filled their values with 10 or -10, it is not clear for the readers.

> Since the highest change score is 9.2 (for betaine), we decided to use 10 or -10 to fill the missing value. 

Page 3, line 77-78: what would be the differences between the metabolite annotation method in their former publication and using METLIN in this manuscript?

> I guess METLIN was updated from the analysis for former publication (it was analyzed 5 years ago), but I do not know exactly.  Anyway, I omitted the reanalyzed data by using METLIN in the first edition of the manuscript.

Page 6, part of "integration pathway analysis": the authors did not describe how they did the integration analysis in detail. For instance, did they upload differentially expressed genes (DEG) and significantly changed metabolites at the meantime, and obtained the enriched pathways and biological processes? OR they upload gene and metabolite lists separately, and then compared the enriched pathways after that?

> I uploaded both DEG list and metabolite list at the meantime in Metaboanalyst database.  I added this explanation. 

The authors may plot a heatmap with DEG and significantly changed metabolites to show the different patterns in control and taurine deficient mice group.

>I added the heatmaps with DEG and changed metabolites for each KEGG map.

Correlation analysis can be a good method for the integration analysis of metabolomics and transcriptomics data.

> I guess this integrated analysis using MetaboAnalyst is a sort of correlation analysis. 

It is good that the authors pointed out in the conclusion that "some of the changes may be directly caused by taurine depletion, while others may be caused by cardiac disorders". However, how to distinguish the direct effects and the secondary effects? It is understandable that it is beyond the scope of this manuscript to distinguish these two effects by conducting additional experiment. But the authors can provide some possible insights and assumptions.

>It is difficult to distinguish the direct effects and the secondary effects.  I believe we can identify the direct effects by comparing with the expression changes in taurine-treated cells/animals. 

The authors should give more details how Fig 2-5 were made? Were they made by Cytoscape? It was not stated in the part of Materials and Methods.

> These map was made by the integrated analysis tool of MetaboAnalyst. This explanation was added in Material and Methods.

The part of metabolite annotation/identification was not satisfying. Because there are four different confidence levels for metabolite annotation. Only with accurate mass for metabolite annotation is the most superficial layer, where the authors must claim this in a very clear way, e.g. state that their annotation was tentative annotations. In addition, the authors still need to clarify the main differences from their last reports.

> I agree that the part of metabolite annotation/identification was not satisfying. Therefore, we decided to omit new identified metabolites from the present analysis. 

Minor comments:

Page 4, line 83: the sentence is broken, not finished.

> This is an omitted part.

The second sentence in Abstract has redundant meanings

> The second sentence was removed. 

Abstract, line 23-24: no need to use quotes for pathways

Page 4, line 91-92: the same like above, no need to use quotes for specific biological processes

>I removed quotes.

Reviewer 3 Report

Review comments for the article entitled “The pathway analysis oftransptomeand metabolite profile to elucidate compensatory mechanism fortaurinedeficiency in the heart oftaurinetransporter knockout mice” by Ito et al.

General Comments:

Ito et al.evaluatedthetranscriptomeand metabolite profiles in thetaurinedeficient heart of its transporter knockout mice. Because this study was carried out using thepreviousreported data, many parts ofmanuscriptwere hard to understand without the previous data. In addition, reviewer confused which the data (results) mentioned in the text body were obtained from the present or previous studies. Furthermore, metabolites and genes in Figures were hard to beidentify, because the symbols and illustrations of KEGG pathway were used as intact. 

Specific comments;

Line 33): Addcomma(,) between “acids”and “etc[1]”.

Lines 44-47): Previous study referred as #8 by Ito et al showed the changes ofacyl- and acetyl-carnitinesin the KO mouse, but there was no data about free-carnitine. 

Lines 44-47): Inprevious studyby Ito et al, the GPC level was increased, but not significantly, in the KO mouse.

Line 47): Glycerophosphocholine (GPC) has been already abbreviated in line 33.

In method section) Data that have been reported in the previous studies were used for the analyses in this study, but the methods how the data were obtained should be observed, at least inbrief explains, for example, the information of experimental animals (number, age, gender, etc.pergroup), the methods of sample treatments for LC-MS analysis, and etc...  

Line 64): One of the commas (,,) must be omitted.

Lines 81-83): Does MWmeanmolecular weight? Please, check the molecular weight value of the compounds again. Add to FC and p-value information ofhydroxyisovalerycarnitine. 

Lines 79-84): The four new identified metabolites could not be measured in the previous LC-MS-based metabolomics analytic study (#8)?

Lines 91-94): Reconsider the English in this sentence.

Lines 91-94, and Table 3): GO cluster analysis showed that the genes ofsymporteractivity and aminoacidstransmembrane transporter activity were significantly increased in terms of metabolism. However, only eleven genes in thesymporterwere listed up in Table3, but the 5 genes in the aminoacidstransmembrane transporter were not shown. In addition, the genes of the catalytic activity that were significantly decreased in the GO cluster analysis werenot also shown.

Lines 102-115 and Fig.1): Reviewer recognizes that the over 1 of M-value (2-fold change) shows the (significant) difference in the gene expression of the KO mouse. It agrees that the genes marked in red,which wereshowedas significance changeby the GO cluster analysis, were plotted around 1 of M-value. However,authorsmentioned that the slc6a12 gene expression is low and is not difference between the KO and WT mice, although the slc6a12 gene marked in blue was plotted at around 1 of M-value.

In Figures 2-5): Authors were explaining the alterations of metabolites and gene in severalmetabolismby applying the KEGG metabolism pathways. It wanders whether the KEGG pathways can be applied to the metabolism specific in heart. In Figures 2-5): Each metabolite and gene could not be identified, because information of KEGG ID were not shown and the size of symbols was too small to read. Information of metabolites can be easily obtained in the KEGG pathway on the internet, but the figures in the present study do not link the network.

In Figures 2-5): Do the fold changes of gene expressions between -2 and 2 mean the significant difference?

Line 139 and Fig.2): Where is the CDP-choline in the Fig.2?

Fig.2) What is the “Choe”?

Lines 134-143): Authors did not mention about the decreased genes showed in green of Fig.2. It wonders whether these decreased genes, especially in C0030, were related to the glycerophospholipid metabolism.

Lines 150-153): The gene expression of Asl was increased as 1.7-fold in the KO mouse, while the metabolites in this pathway, especially in arginosuccinate that is catalyzed directly by this enzyme, were also increased. What is the reason on this discrepancy? Likewise, the Acat1, Acca2, Bdh1, and Mccc2 that metabolize directly/indirectly acetyl-CoA and/or propionyl-CoA in fatty acid/BCAA metabolism were decreased, while acetyl-CoA and propionyl-CoA were increased, in Fig.4a/b and Fig.5.

Lines 153-154): In the ornithine cycle in liver, arginosuccinate that is formed from citrulline and aspartate is metabolized to arginine and fumarate by Asl. However, authors mentioned that “In the case of other tissues, Asl catalyzes the formation of arginine from citrulline and arginosuccinate”. Is it true that arginine is metabolized from both citrulline and arginosuccinate in the other tissues such as heart? 

Lines 154-159): Authors considered the alterations of metabolite levels and gene expressions in the arginine and alanine metabolism, particularly in Asl, in relation to the NOS generation in the KO mouse. Have the authors confirmed previously whether the NOS generation was increased in the heart of the KO mouse?

Lines 168-170): Authors mentioned that short-chain carnitines (C2, C4, C6, C8) are higher in the KO mouse. Where the results shown? 

Lines 170-173): Carnitine plays as carrier of long-chain fatty acids to cross the mitochondrial membrane, but both medium- and short-chain fatty acids do not need carnitine to cross the mitochondrial membrane. Therefore, it wonders whether the increased acyl-carnitines (C4-C8) mean the incomplete fatty acid oxidation.

Line 175, and Fig.4B): Which are correct, Acot in the text and Acat1 in Fig.4B?

Lines 174-176): As mentioned the above, the gene expressions of Bdh1 and Acat1 (Acot?) that metabolize to acetyl-CoA from ketone bodies were decreased, but acetyl-CoA was increased.

Line 184): Exchange “Valine” to “valine”.

Lines 185-194): Valine is metabolized to propionyl-CoA, leucine is metabolized to acetyl-CoA, and isoleucine is metabolized to both propionyl-CoA and acetyl-CoA. It agrees that leucine and isoleucine catabolism might be suppressed due to decreased gene expressions of Mccc2 and Acaa2, respectively, and in turn, hydroxyisovaleryl-CoA and 2-methylbutanoyl-CoA were increased, and consequently, converted to hydroxyisovalerylcarnitine and 2-methylbutanoylcarnitine, respectively. However, the data suggest that the catabolism of valine to propionyl-CoA might be increased, but the reason is unclear why propionylcarnitine was increased. Although the downstream of propionyl-CoA catabolism illustrated in Fig.5 did not show, there is a possibility that propionyl-CoA catabolism might be also decreased? 

Line 215): Add “Slc” in front of “38a4”.

Line 206): Change “Taurine transporter” to “TauT”, because authors defined the transporter name in Line 36. Several names of Slc6a12 were written such as betaine/GABA transporter-1 and GABA transporter-1. Like “TauT”, use the defined name “BGT-1”. 

Lines 116-117, 223-228): In addition to OCTN1, OCTN2 is also one of specific carnitine transporter in heart. Have authors observed the expression of OCTN2 in the TauTKO heart? In addition to the comment for lines 44-47, OCTNs that are the cytoplasm transmembrane transporter are carrier of free-carntine, but not acyl- and acetyl-carnitines that were significantly increased in the TauTKO heart reported by the previous study #8. Do authors want to focus on the free-carntine level (regulation of osmolytes) rather than the increased levels of acyl/acetyl-carnitine (beta-oxidation metabolism)?

Lines 229-231): The explained metabolic pathways are hard to understand in the indicated KEGG pathway in Fig.2, because of small characters and specific IDs.

Lines 233-234): Where are the result data indicating that Pla2g9 and Ache genes were induced and PLB gene was not induced in TauTKO heart in the present study?

Line 237): The sentence “the letter which…..” is unclear.

Line 239): Which are “phosphatidic acid”, “diacylglycerol” and “phosphatidylcholine” in Fig.2? 

Lines 265-267): Authors suggested that fatty acid beta-oxidation might be suppressed (authors expressed as “slow”) by the feedback of acetyl-CoA accumulation. If the beta-oxidation were decreased, why the acetyl-CoA was increased? If the reaction of beta-oxidation were inhibited to slow in response to the catabolic speed of acetyl-CoA, the acetyl-CoA should not be accumulated. Where the acetyl-CoA come from? How do authors explain the increase of acyl-carnitines as well as acetyl-carnitine?

Lines 271-277): Previous study has reported that the inactivation of BCAA catabolism by knockout of BCKAs dehydrogenase caused impaired cardio function, and also that pressure overload increased BCKAs levels. However, some enzymes that are the downstream of BCAA catabolic pathway were decrease, but the genes of BCKAs dehydrogenases were not changed in the TauTKO shown in Fig.5. Have authors reported the changes of BCKAs in the TauTKO heart? In the present study, BCAAs were not contained in the significantly changed metabolites in the TauTKO heart showed in Table 1. In addition, there are the previous studies showing the role of taurine on BCAA metabolism?

Author Response

> First, I appreciate the reviewers for his/her very helpful and careful review for our paper. We have revised the manuscript according to the reviewer’s comments. We believe that this paper is improved well. 

The response to the comments is as follows;

(Additional descriptions according to reviewers’ comments are lined by yellow marker (reviewer 1) and light-green marker (reviewer 2), light-blue marker (reviewer 3))

General Comments:

Ito et al.evaluatedthetranscriptomeand metabolite profiles in thetaurinedeficient heart of its transporter knockout mice. Because this study was carried out using thepreviousreported data, many parts ofmanuscriptwere hard to understand without the previous data. In addition, reviewer confused which the data (results) mentioned in the text body were obtained from the present or previous studies. Furthermore, metabolites and genes in Figures were hard to beidentify, because the symbols and illustrations of KEGG pathway were used as intact.

Specific comments;

Line 33): Addcomma(,) between “acids”and “etc[1]”.

> I added a comma.

Lines 44-47): Previous study referred as #8 by Ito et al showed the changes of acyl- and acetyl-carnitines in the KO mouse, but there was no data about free-carnitine.

> Since L-carnitine is also significantly changed in Metabolome analysis (p<0.05 by Mann-Whitney U test), we added it in the list used for pathway analysis.  Since the VIP score calculated by PLS-DA was 0.9 for carnitine, it was not listed in the previous manuscript.  I added the explanation about the criteria (VIP is greater than 1 or P value calculated by Mann-Whitney test is less than 0.05) in the method section.

Lines 44-47): Inprevious studyby Ito et al, the GPC level was increased, but not significantly, in the KO mouse.

> VIP score for GPC calculated from PLS-DA is greater than 1.  Since value of VIP score which is greater than 1 is generally used as criteria for variable selecton, we analyzed the metabolites which VIP score is greater than 1.   

Line 47): Glycerophosphocholine (GPC) has been already abbreviated in line 33.

> Thank you for the careful reading. I changed.

In method section) Data that have been reported in the previous studies were used for the analyses in this study, but the methods how the data were obtained should be observed, at least inbrief explains, for example, the information of experimental animals (number, age, gender, etc.pergroup), the methods of sample treatments for LC-MS analysis, and etc... 

> The brief explains were added in the method section. 

Line 64): One of the commas (,,) must be omitted.

> I omitted a comma.

Lines 81-83): Does MWmeanmolecular weight? Please, check the molecular weight value of the compounds again. Add to FC and p-value information ofhydroxyisovalerycarnitine.

> I am sorry for confusing the reviewer but I showed the weight for M+H+ instead of molecular weight for metabolites.  We decided to omit this section and use of 4 metabolites in this study.  It is because compound data obtained from METLIN is putative and we actually could not have confidence that these unknown compounds are Lysophosphatidylcholine, Methylbutanoylcarnitine and Hydroxyisovalerylcarnitine, as the reviewer #3 pointed.  Concerning propionylcarnitine, it was already annotated (I could not notice about it when I prepared the previous manuscript). Since this compound satisfied the criteria (p<0.05), it is listed in Table 1.  Then, integrated analysis was performed on this data set.

Lines 79-84): The four new identified metabolites could not be measured in the previous LC-MS-based metabolomics analytic study (#8)?

> Since the data in the pervious report was analyzed 5 years ago, the database may be updated. I guess this is a reason why we could annotate the new metabolites in the present study.  However, as pointed by Reviewer #3, the reanalysis by using METLIN only is the most superficial layer. So we omitted this section. 

Lines 91-94): Reconsider the English in this sentence.

> I modified the sentence. 

Lines 91-94, and Table 3): GO cluster analysis showed that the genes ofsymporteractivity and aminoacidstransmembrane transporter activity were significantly increased in terms of metabolism. However, only eleven genes in the symporter were listed up in Table3, but the 5 genes in the amino acids transmembrane transporter were not shown. In addition, the genes of the catalytic activity that were significantly decreased in the GO cluster analysis were not also shown.

> We added more two tables for the amino acids transmembrane transporter and the genes of the catalytic activity.

Lines 102-115 and Fig.1): Reviewer recognizes that the over 1 of M-value (2-fold change) shows the (significant) difference in the gene expression of the KO mouse. It agrees that the genes marked in red,which wereshowedas significance changeby the GO cluster analysis, were plotted around 1 of M-value. However,authorsmentioned that the slc6a12 gene expression is low and is not difference between the KO and WT mice, although the slc6a12 gene marked in blue was plotted at around 1 of M-value.

> The change for slc6a12 is not statistically significant.  I added the explanation. 

In Figures 2-5): Authors were explaining the alterations of metabolites and gene in several metabolism by applying the KEGG metabolism pathways. It wanders whether the KEGG pathways can be applied to the metabolism specific in heart. In Figures 2-5): Each metabolite and gene could not be identified, because information of KEGG ID were not shown and the size of symbols was too small to read. Information of metabolites can be easily obtained in the KEGG pathway on the internet, but the figures in the present study do not link the network.

> As the reviewer suggested, KEGG pathway is difficult to apply to the metabolism specific in heart; for example, many of genes are not expressed in the heart.  I guess that this is a reason why we could hardly detect metabolic pathways which are statistically significant in the integrated analysis. 

Also according to the reviewers comment, I modified the KEGG pathway maps to be easy to read.    

In Figures 2-5): Do the fold changes of gene expressions between -2 and 2 mean the significant difference?

> As described in method section, we recognized the fold change more than 1.5 or less than -1.5 is as significant difference.

Line 139 and Fig.2): Where is the CDP-choline in the Fig.2?

> Metabolite names including CDP-choline were added in Figures.

Fig.2) What is the “Choe”?

> “Choe” is wrong.  I meant “Ache”.

Lines 134-143): Authors did not mention about the decreased genes showed in green of Fig.2. It wonders whether these decreased genes, especially in C0030, were related to the glycerophospholipid metabolism.

> C00930 is Glycerol-1-phosphate. The reduction of this metabolite may be caused by enhancement of phosphatidylcholine production.  Discussion about the decrease of this metabolite was added.

Lines 150-153): The gene expression of Asl was increased as 1.7-fold in the KO mouse, while the metabolites in this pathway, especially in arginosuccinate that is catalyzed directly by this enzyme, were also increased. What is the reason on this discrepancy? Likewise, the Acat1, Acca2, Bdh1, and Mccc2 that metabolize directly/indirectly acetyl-CoA and/or propionyl-CoA in fatty acid/BCAA metabolism were decreased, while acetyl-CoA and propionyl-CoA were increased, in Fig.4a/b and Fig.5.

> The reason on the discrepancy between Asl and arginosuccinate may be caused by the increase in citrulline and ornithine. The explanation for this discrepancy was added in result section.     

Lines 153-154): In the ornithine cycle in liver, arginosuccinate that is formed from citrulline and aspartate is metabolized to arginine and fumarate by Asl. However, authors mentioned that “In the case of other tissues, Asl catalyzes the formation of arginine from citrulline and arginosuccinate”. Is it true that arginine is metabolized from both citrulline and arginosuccinate in the other tissues such as heart?

> I am sorry for confusing the reviewer.  Arginine is metabolized only from arginosuccinate. 

Lines 154-159): Authors considered the alterations of metabolite levels and gene expressions in the arginine and alanine metabolism, particularly in Asl, in relation to the NOS generation in the KO mouse. Have the authors confirmed previously whether the NOS generation was increased in the heart of the KO mouse?

>No, we did not check NOS generation in TauTKO mice. 

Lines 168-170): Authors mentioned that short-chain carnitines (C2, C4, C6, C8) are higher in the KO mouse. Where the results shown?

> The increases in short-chain carnitines were previously reported in reference #8.  I added the reference.

Lines 170-173): Carnitine plays as carrier of long-chain fatty acids to cross the mitochondrial membrane, but both medium- and short-chain fatty acids do not need carnitine to cross the mitochondrial membrane. Therefore, it wonders whether the increased acyl-carnitines (C4-C8) mean the incomplete fatty acid oxidation.

> Carnitine is responsible not only for translocation of acyl-CoA from cytosol to mitochondria but also for detoxification of excess acyl-CoA. For example, butyrylcarnitine is increased in the plasma of the patients of Short-chain acyl-CoA dehydrogenase deficiency. This explanation is added in the result section.

Line 175, and Fig.4B): Which are correct, Acot in the text and Acat1 in Fig.4B?

> Acat1 is correct.

Lines 174-176): As mentioned the above, the gene expressions of Bdh1 and Acat1 (Acot?) that metabolize to acetyl-CoA from ketone bodies were decreased, but acetyl-CoA was increased.

> It is possible that the accumulated acetyl-CoA may suppress the expression of several genes which is responsible for acetyl-CoA production as negative feedback.  I added this discussion in discussion section.

Line 184): Exchange “Valine” to “valine”.

> I exchanged.

Lines 185-194): Valine is metabolized to propionyl-CoA, leucine is metabolized to acetyl-CoA, and isoleucine is metabolized to both propionyl-CoA and acetyl-CoA. It agrees that leucine and isoleucine catabolism might be suppressed due to decreased gene expressions of Mccc2 and Acaa2, respectively, and in turn, hydroxyisovaleryl-CoA and 2-methylbutanoyl-CoA were increased, and consequently, converted to hydroxyisovalerylcarnitine and 2-methylbutanoylcarnitine, respectively. However, the data suggest that the catabolism of valine to propionyl-CoA might be increased, but the reason is unclear why propionylcarnitine was increased. Although the downstream of propionyl-CoA catabolism illustrated in Fig.5 did not show, there is a possibility that propionyl-CoA catabolism might be also decreased?

> I have no idea whether propionyl-CoA catabolism is decreased, but it is possible.  But, as I described, I guess that a reduction in Abat and Aldh7a1 may suppress the metabolism of (S)-methylmalonate semialdehyde (C06002), resulting in enhanced generation of propionyl CoA (C00100) conversion from (S)-methylmalonate semialdehyde.

Line 215): Add “Slc” in front of “38a4”.

>I added.

Line 206): Change “Taurine transporter” to “TauT”, because authors defined the transporter name in Line 36. Several names of Slc6a12 were written such as betaine/GABA transporter-1 and GABA transporter-1. Like “TauT”, use the defined name “BGT-1”.

> I exchanged to TauT and BGT-1.

 Lines 116-117, 223-228): In addition to OCTN1, OCTN2 is also one of specific carnitine transporter in heart. Have authors observed the expression of OCTN2 in the TauTKO heart? In addition to the comment for lines 44-47, OCTNs that are the cytoplasm transmembrane transporter are carrier of free-carntine, but not acyl- and acetyl-carnitines that were significantly increased in the TauTKO heart reported by the previous study #8. Do authors want to focus on the free-carntine level (regulation of osmolytes) rather than the increased levels of acyl/acetyl-carnitine (beta-oxidation metabolism)?

> Both OCTN1 (slc22a4) and OCTN2(slc22a5) are not different.  I added the plot and text for OCTN2. Concerning carnitine induction, we focused on the free carnitine level as osmolyte regulation. 

Lines 229-231): The explained metabolic pathways are hard to understand in the indicated KEGG pathway in Fig.2, because of small characters and specific IDs.

> I totally changed metabolism map to understand easily.

Lines 233-234): Where are the result data indicating that Pla2g9 and Ache genes were induced and PLB gene was not induced in TauTKO heart in the present study?

> Pla2g9 is wrong.  Pla2a4 is correct.  As revewer#2 commented, I added the heatmap to show different expressed genes and metabolites. 

Line 237): The sentence “the letter which…..” is unclear.

> I rewrote this sentence. 

Line 239): Which are “phosphatidic acid”, “diacylglycerol” and “phosphatidylcholine” in Fig.2?

> These metabolites are also named on the map in Fig. 2. 

Lines 265-267): Authors suggested that fatty acid beta-oxidation might be suppressed (authors expressed as “slow”) by the feedback of acetyl-CoA accumulation. If the beta-oxidation were decreased, why the acetyl-CoA was increased? If the reaction of beta-oxidation were inhibited to slow in response to the catabolic speed of acetyl-CoA, the acetyl-CoA should not be accumulated. Where the acetyl-CoA come from? How do authors explain the increase of acyl-carnitines as well as acetyl-carnitine?

> It is possible that the accumulated acetyl-CoA is come from Glycolysis activation and/or the reduction of TCA cycle.  Indeed, we previously observed that the glycolysis is enhanced and the TCA cycle is diminished in the taurine-deficient heart. 

Lines 271-277): Previous study has reported that the inactivation of BCAA catabolism by knockout of BCKAs dehydrogenase caused impaired cardio function, and also that pressure overload increased BCKAs levels. However, some enzymes that are the downstream of BCAA catabolic pathway were decrease, but the genes of BCKAs dehydrogenases were not changed in the TauTKO shown in Fig.5. Have authors reported the changes of BCKAs in the TauTKO heart? In the present study, BCAAs were not contained in the significantly changed metabolites in the TauTKO heart showed in Table 1. In addition, there are the previous studies showing the role of taurine on BCAA metabolism?

> No, we did not check BCKAs in TauTKO hearts.  Also we did not know the role of taurine on BCAA metabolism. 

Round 2

Reviewer 1 Report

The manuscript has been improved tremendously and certain parts that were not fully explained are now adequately explained.

Reviewer 2 Report

I think the revised version can be accepted.

Reviewer 3 Report

Most ofquestionswere solved by the authors in the revised version.